# Fragile protein folds: Sequence and environmental factors affecting the equilibrium of two interconverting, stably folded protein conformations

Xingjian Xu[1,2], Igor Dikiy[1,3], Matthew R. Evans[4], Leandro P. Marcelino[1,5], Kevin H. Gardner[1,5,6]

[1]Structural Biology Initiative, CUNY Advanced Science Research Center, New York, NY, USA

[2]Ph.D. Program in Biochemistry, The Graduate Center, CUNY, New York, NY, USA

[3]Current address: Regeneron Pharmaceuticals, Tarrytown, NY, USA

[4]Current address: Acclaim Physician Group, Inc. Fort Worth, TX, USA

[5]Department of Chemistry and Biochemistry, City College of New York, New York, NY, USA

[6]Biochemistry, Chemistry and Biology Ph.D. Programs, The Graduate Center, CUNY, New York, NY, USA

**Correspondence:** Kevin Gardner (kgardner@gc.cuny.edu)

**Keywords**: metamorphic proteins, NMR, Per-ARNT-Sim, protein folding, ligand binding; ARNT

**Abstract.** Recent research on fold-switching metamorphic proteins has revealed some notable exceptions to Anfinsen's hypothesis of protein folding. We have previously described how a single point mutation can enable a well-folded protein domain, one of the two PAS (Per-ARNT-Sim) domains of the human ARNT (aryl hydrocarbon receptor nuclear translocator) protein, to interconvert between two conformers related by a slip of an internal β-strand. Using this protein as a test case, we advance the concept of a "fragile fold," a protein fold that can reversibly rearrange into another fold that differs by a substantial number of hydrogen bonds, entailing reorganization of single secondary structure elements to more drastic changes seen in metamorphic proteins. Here we use a battery of biophysical tests to examine several factors affecting the equilibrium between the two conformations of the switching ARNT PAS-B Y456T protein. Of note, we find that factors which impact the HI loop preceding the shifted Iβ-strand affect both the equilibrium levels of the two conformers and the denatured state which links them in the interconversion process. Finally, we describe small molecules that selectively bind to and stabilize the wildtype conformation of ARNT PAS-B. These studies form a toolkit for studying fragile protein folds and could enable ways to modulate the biological functions of such fragile folds, both in natural and engineered proteins.

# 1 Introduction

Anfinsen's hypothesis, which states that a protein's primary sequence encodes a unique fold or conformation, has dominated the study of protein folding for almost 50 years (Anfinsen, 1973). However, it is increasingly clear that a range of exceptions to the "one sequence, one fold" concept are widely found in biology. One counterexample is the intrinsic disorder found in a significant number of functional proteins and protein regions. These intrinsically-disordered regions do not adopt a stable three-dimensional structure, instead existing as conformational ensembles of states that may include pre-formed structural nuclei (Dyson, 2016). Other counterexamples are provided by proteins which interconvert among multiple folded states, ranging from a "fragile fold", in which substitution of a few amino acids – even one – results in the domain co-existing in two states (Evans et al., 2009;Evans and Gardner, 2009;Ha and Loh, 2012), to the general concept of a "metamorphic protein," one that can reversibly adopt different stable folds in different environmental conditions (Murzin, 2008).

We consider a protein or domain to have a "fragile fold" if it can populate two stable folds and interconvert reversibly between them, doing so via breaking and reforming a substantial number of hydrogen bonds. This concept differs from that of simple protein conformational switches (Gerstein and Echols, 2004) mainly in the extent of hydrogen bond network remodeling. For example, we previously reported the Y456T mutant of PAS-B (Per-ARNT-Sim) domain of the human ARNT (Aryl hydrocarbon Receptor Nuclear Translocator) protein, which undergoes a 3-residue β-strand slip requiring 15 of the 26 inter-strand hydrogen bonds in the β-sheet to be broken and re-formed while still retaining the same overall topology (Evans et al., 2009;Evans and Gardner, 2009). In metamorphic proteins, which can be considered an extreme case of a fragile fold, an even greater number of hydrogen bonds differ between the two folds as seen in lymphotactin (which alternates between a conventional chemokine fold and a dimeric β-sandwich (Kuloğlu et al., 2002;Tuinstra et al., 2008)). These studies suggest that protein folds may hide unforeseen flexibility or fragility which can be trivially accessed by small changes in sequence or environmental conditions, facilitating the evolution of new folds and protein domains (Yadid et al., 2010;Tuinstra et al., 2008;Alexander et al., 2009;Dishman et al., 2021).

To experimentally probe such fragile folds, we here use the abovementioned ARNT PAS-B Y456T point mutant as a model system. ARNT is a eukaryotic bHLH-PAS (basic-helix-loop-helix – Per-ARNT-Sim) transcription factor that dimerizes with other bHLH-PAS proteins (e.g. AhR, HIF-α) to bind DNA and regulate gene expression in response to a varied set of stimuli (Labrecque et al., 2013). In addition to the bHLH DNA binding module and C-terminal coactivator domains, these proteins contain two PAS domains (PAS-A and PAS-B), which are essential for heterodimerization (Erbel et al., 2003;Chapman-Smith et al., 2004;Wu et al., 2015). The typical PAS domain fold is 100-120 residues in length and consists of several α-helices packed against one side of a five-stranded, anti-parallel β-sheet, often enclosing a ligand-binding cavity (Card et al., 2005;Wu et al., 2015;Wu et al., 2019;Bisson et al., 2009), **Figure 1a**. PAS domains provide binding sites for many diverse binding partners, including other PAS domains (Cardoso et al., 2012;Erbel et al., 2003;Huang et al., 2012;Scheuermann et al., 2009;Wu et al., 2015), adjacent helices (Harper et al., 2003;Nash et al., 2011;Rivera-Cancel

et al., 2014), and coiled-coil coactivators (Guo et al., 2013;Guo et al., 2015). Notably, changes in the occupancy or configuration of bound small molecule ligands can modulate the protein/protein interactions of many PAS domains (Scheuermann et al., 2009), opening the door to natural and artificial control of signaling pathways.

While designing point mutants to disrupt the interactions between the ARNT PAS-B domain and other binding partners, we fortuitously discovered that the Y456T variant existed in a slow conformational equilibrium between two equally-populated conformations (Evans et al., 2009). Solution NMR studies revealed that the new conformation primarily differs from the native fold by a +3 slip in register and accompanying inversion of the central Iβ-strand (**Figure 1b**); hence, we termed the conformations "WT" and "SLIP." Interestingly, the SLIP conformation does not bind HIF-2α PAS-B (Evans et al., 2009), consistent with the involvement of the ARNT PAS-B beta sheet in this interaction (Scheuermann et al.,

2009;Wu et al., 2015). We subsequently discovered 105 $\mathrm{\AA}^3$ of interconnected internal cavities in a high-resolution WT crystal structure (Guo et al., 2013), which are likely to have collapsed in the SLIP conformation (**Figure 1b**) (Xu et al., 2021). Additionally, interconversion between the two conformations appears to require global unfolding, as revealed by data from a combination of solution NMR approaches, spanning real-time NMR (Evans and Gardner, 2009) to high-pressure perturbation of the equilibrium (Xu et al., 2021) and as might be expected from the large number of hydrogen bonds

disrupted by the beta-strand slip (**Supplementary Figure S1**).

Here we more broadly examine other factors that determine the equilibrium populations of these two folds, with interest in both characterizing these effects and artificially manipulating the interconversion as a possible route to engineering novel switchable proteins. We start by characterizing the temperature dependence of the interconversion process, allowing us to extract thermodynamic parameters of it. We also show that several features of the HI loop region,

which precedes the shifting Iβ strand, can also influence the WT:SLIP ratio. In addition, we demonstrate mutations in this region and the Hβ-strand proceeding the HI loop can also affect the denatured state of the protein, resulting in differential refolding to one of the two conformations during interconversion. Lastly, we identify compounds that can preferentially bind one conformer over another, giving us the ability to shift the relative populations between these two conformations. Many of the approaches discussed here exemplify ways of studying and manipulating other fragile folds or metamorphic proteins,

both in natural and engineered contexts as ligand-regulated sensors and switches (Ha and Loh, 2017).

## 2      Materials and Methods

### 2.1      Cloning, expression, and purification of ARNT PAS-B mutants

Plasmid DNA encoding the human ARNT PAS-B domain [residues 356-470] was used to introduce various point mutations, using the QuikChange Site-Directed Mutagenesis Kit (Stratagene). Following PCR amplification and digestion of

the parental template with DpnI, using the manufacturers' instructions, the modified plasmid was transformed into the pHis-parallel bacterial expression vector (Sheffield et al., 1999). The accuracy of each mutation was verified by sequencing.

ARNT PAS-B mutant constructs were transformed into *E. coli* BL21(DE3) cells. For isotopically-labelled protein expression, cultures were grown in M9 minimal media containing 1 g/L $^{15}$NH$_4$Cl for uniformly $^{15}$N-labeled samples or 1 g/L $^{15}$NH$_4$Cl and 3 g/L $^{13}$C glucose for uniformly $^{13}$C,$^{15}$N-labeled samples, at 37 °C. Once the cell density (A$_{600}$) measured 0.6-0.8, protein expression was induced with 0.5 mM isopropyl β-D-thiogalactoside. Following 15 hr at 20 °C, cell pellets were harvested and resuspended in 20 mL of 50 mM Tris (pH 7.5), 15 mM NaCl and 20 mM imidazole. Cells were lysed by high-pressure extrusion, centrifuged, and filtered using a 0.45 μm pore size filter. The supernatant was loaded over a Ni$^{2+}$-NTA affinity column and eluted using a linear 20-500 mM imidazole gradient. The resulting sample was exchanged into an imidazole-free buffer (50 mM Tris [pH 7.5], 15 mM NaCl) and incubated overnight in the presence of His$_6$-TEV protease. Following His$_6$-tag cleavage, the remaining protein was purified away from free His$_6$-tag and His$_6$-TEV with an additional pass over a Ni$^{2+}$-NTA column and concentrated in an Amicon pressure-driven ultrafiltration cell (MilliporeSigma, Burlington, MA) with YM-10 10 kDa filters. The resulting protein contains a four-residue vector-derived N-terminal cloning artifact, GAMD, plus residues 356-470 of ARNT PAS-B.

## 2.2 NMR analysis of ARNT PAS-B Y456T equilibrium constant

Solution NMR studies of ARNT PAS-B and its mutants were conducted in 50 mM Tris, 17 mM NaCl and 5 mM DTT. To determine the effects that buffer and pH have on the stability of each conformation, ARNT PAS-B Y456T was exchanged into either 50 mM Tris (pH range 7-9) or PIPES (pH range 6-7.5), 17 mM NaCl, and 5 mM DTT. $^{15}$N/$^1$H HSQC spectra of 200 μM protein were recorded more than 20 hr post-exchange to allow adequate time for any perturbations to the conformational equilibrium to be established. To determine the effects of temperature on the equilibrium constant, $^{15}$N/$^1$H HSQC spectra of 200 μM ARNT PAS-B Y456T in 50 mM Tris, 17 mM NaCl and 5 mM DTT were recorded after incubation for a sufficient time (2-4 hr) to ensure equilibrium was established at temperatures between 278 K and 333 K in 5 K increments. All NMR spectra were acquired on Varian Inova (Palo Alto, CA) and Bruker Avance III (Billerica, MA) spectrometers, processed using NMRpipe (Delaglio et al., 1995) and NMRFx (Johnson, 2018;Norris et al., 2016) and analyzed using NMRViewJ (Johnson and Blevins, 1994;Johnson, 2018). We measured the WT:SLIP conformational equilibrium as the relative volume of several well-dispersed peaks originating from each conformation.

## 2.3 NMR analysis of unfolded state of ARNT PAS-B mutants

We populated the unfolded state of the wildtype, Y456T, and F444Q/F446A/Y456T ARNT PAS-B proteins by urea denaturation. HNCO, HN(CA)CO, HNCACB, and CBCA(CO)NH spectra were collected at 25 °C on 500 μM uniformly $^{13}$C,$^{15}$N-labeled protein in 8 M urea, 50 mM Tris pH 7.5, 20 mM NaCl, 5 mM beta-mercaptoethanol and used to assign ~95% of the HN, N, C', Cα, and Cβ resonances. We then used HN, N, Cα, and Cβ chemical shifts to predict S$^2$ order parameters using the TALOS+ program (Shen et al., 2009) and the Cα and Cβ chemical shifts to predict residual secondary structure

using the SSP program (Marsh et al., 2006). Chemical shift assignments for the urea-denatured states of the proteins have been submitted to the BMRB database as noted in **Data Availability**.

**2.4     NMR-based evaluation of small molecule binding to ARNT PAS-B WT and F444Q/F446A/Y456T variant**

Small molecule ligands for ARNT PAS-B wildtype protein were identified in a previously-reported NMR-based fragment screen (Guo et al., 2013). In summary, this screen utilized $^{15}$N/$^{1}$H HSQC spectra to monitor ligand binding as overviewed in **Figure 4a**, based on comparisons of spectra of $^{15}$N-labeled ARNT PAS-B wildtype recorded in isolation and in the presence of potential small molecule ligands. We assembled a library of 762 chemical fragments (average MW 203

130 Da, ±73 Da s.d.), initially screening these in pools of five (250 μM protein, 1 mM ligand), with hits identified as pools which produced substantial changes in $^{15}$N/$^{1}$H peak locations or intensities. Compounds from such pools were subsequently individually screened with single point (500 μM) additions to $^{15}$N-labeled ARNT PAS-B, with individual hits titrated over six or more concentrations to establish initial estimates of affinity and binding site location. As noted in (Guo et al., 2013), ten compounds from this screen were considered to be ARNT PAS-B binders; data supporting this are shown in titrations

shown in **Figure 4** (KG-548) and **Supplementary Figure S7**. All ten of these compounds were then counterscreened against the F444Q/F446A/Y456T ARNT PAS-B variant, with $^{15}$N/$^{1}$H HSQC spectra recorded of 360 μM $^{15}$N-labeled protein and 500 μM ligand.

**2.5     KG-548 and KG-655 titration analyses**

To determine the binding affinity for compound KG-548, we conducted a $^{15}$N/$^{1}$H HSQC titration series with

140 increasing concentrations of KG-548 (0, 25, 50, 100, 200, 500, and 1000 μM) against a constant 200 μM concentration of ARNT PAS-B Y456T. $^{15}$N/$^{1}$H HSQC spectra were recorded 24 hr after samples were mixed to allow sufficient time to reach equilibrium. The concentration of ligand-bound WT conformation ($C_{WT,B}$) was derived from peak intensities, plotted versus the ligand concentration ($C_L$), and fit to the following equation to derive dissociation constant ($K_d$) and maximal binding ($B_{max}$):

$$C_{WT,B} = B_{max} \times C_L/(K_d + C_L) \qquad (1)$$

We attempted to determine the concentration of ligand-bound protein directly from peak intensities; however, this analysis was complicated by the broadening of WT peaks in the presence of KG-548. In order to overcome this challenge, we used the following protocol to determine the concentration of ligand-bound WT conformation.

We measured peak volumes for those peaks that showed no chemical shift changes, which report on both bound and

150 unbound states of the protein. The relative peak volumes of the peaks corresponding to each conformation indicate the relative concentrations of total WT and SLIP conformations, but not the ligand-bound and ligand-free concentrations. We calculated the concentration of the SLIP conformation ($C_{+3}$), which does not bind compound appreciably, and the total concentration of WT conformation ($C_{WT,T}$) at each concentration of compound, by comparing these peak volumes ($V_{WT}$, $V_{+3}$) to those in the absence of compound:

$$C_{WT,T} = C_{WT,0} \times (V_{WT}/V_{WT,0}) \qquad (2)$$

$$C_{+3} = C_{+3,0} \times (V_{+3}/V_{+3,0}) \qquad (3)$$

Then, assuming that the SLIP conformation remains in the same equilibrium with the unbound WT conformation as in the absence of compound, we estimated the concentration of ligand-bound WT conformation as:

$$C_{WT,B} = C_{WT,T} - C_{+3} \qquad (4)$$

Most of the residues used in the titration analysis for KG-548 showed chemical shift changes and peak broadening in the presence of KG-655, rendering them unusable for the method described above. We therefore conducted $^{13}C/^{1}H$ HSQC titration experiments instead. We monitored the L391 $\delta$1 methyl peaks which are particularly upfield shifted and well resolved for both WT and SLIP conformations (Evans and Gardner, 2009). L391 is also a good probe to extract and compare binding affinities of KG-548 and KG-655 as it is not directly involved in the binding of either compounds (and thus has minimal chemical shift changes observed in the presence of KG-548 or KG-655). We conducted the $^{13}C/^{1}H$ HSQC titration series with increasing concentration of KG-655 (0, 500, 1000, 2500, 5000, 10000 μM) against 250 μM concentration of ARNT PAS-B Y456T and extracted $C_L$ and $K_d$ as described above, assuming the equilibrium between the unbound WT conformation and the SLIP conformation remains unchanged at all ligand concentrations. For comparison, we additionally conducted a $^{13}C/^{1}H$ HSQC titration series for KG-548 (0, 500, 1000, 2000, 3000, 4000 μM) against 250 μM concentration of ARNT PAS-B Y456T. We attempted to match the concentration of KG-548 to the concentrations used for KG-655, but were unable to exceed a maximum of 4000 μM for KG-548 without substantially increasing DMSO concentration above 2% DMSO.

## 3        Results

### 3.1        Impact of temperature and ionic strength on conformational equilibrium

To characterize the thermodynamics of the WT:SLIP conformational change, we used $^{15}N/^{1}H$ HSQC spectra of ARNT PAS-B Y456T to calculate the populations of WT and SLIP states at temperatures between 278 K and 328 K (**Figure S2**). Below 300K, the conformational equilibrium is linearly dependent on temperature from 275 to 303 K (**Supplementary Figure 2a**), with a progressive bias towards the WT conformation with increasing temperature, suggesting that the conversion process is endothermic. Using data 303K and below, we plotted the natural logarithm of the equilibrium constant versus the inverse of the temperature and used the linear region of the plot to determine ΔH (6.8 kcal/mol), ΔS (23.9 cal/mol-K), and $\Delta G_{298K}$ (-0.31 kcal/mol) of the SLIP to WT conversion (**Figure 2b**), confirming that it is indeed endothermic but slightly favored overall due to entropic considerations.

Above 300 K, we observed a levelling of the equilibrium at approximately 2:1 (WT:SLIP) (**Supplementary Figure S2b**). We attribute this to protein aggregation at higher temperatures, as seen by a decrease in peak intensity across the

[15]N/[1]H HSQC spectrum, likely due the partial unfolding during interconversion between WT and SLIP conformations (Evans and Gardner, 2009;Xu et al., 2021).

Studies on several proteins that slowly interconvert between two different conformations, such as lymphotactin (Tuinstra et al., 2008) and drk SH3 (Zhang and Forman-Kay, 1995), have shown that solvent conditions such as counterion identity and ionic strength can readily shift such equilibria. Therefore, we determined whether changes in such parameters similarly affect the equilibrium between the two folded states of ARNT PAS-B Y456T. We assessed the impact of changing buffers and pH values (either Tris or PIPES buffers, over total range of pH 6.0 – 9.0), along with salt concentrations (50 – 200 mM NaCl) by using [15]N/[1]H HSQC spectra to determine the relative populations the two states (**Supplementary Figure S3**). We discovered that these effects were minimal, as no substantial (greater than a 5% deviation) changes were observed for independent pH and salt titrations. These results are consistent with the relatively similar amino acid types exposed to solvent in the two conformations of the Iβ-strand.

### 3.2 Impact of HI loop length and sequence on conformational equilibrium

We previously reported that additional mutations in the Y456T background can alter the WT:SLIP equilibrium (**Table 1, left column**). At an extreme, we could lock the protein into the SLIP conformation by adding the F444Q and F446A point mutations on the Hβ strand to the Y456T background, enabling structural characterization of the SLIP conformation (Evans et al., 2009). Another potential contributor to the WT:SLIP equilibrium is the HI loop (residues 447-454), which connects the Hβ and the Iβ strand (**Figure 1a**). This loop is highly flexible in the wildtype ARNT PAS-B domain, showing intermediate exchange broadening of NMR signals at several sites (Card et al., 2005). In the SLIP conformation, the HI loop shortens and packs more tightly against the domain core; as well, the N448-P449 peptide bond within the loop isomerizes from trans (WT) to cis (SLIP). We have previously shown that although the P449 residue has minor impact on the equilibrium WT:SLIP levels, this residue plays a critical role in the kinetics of interconversion (Evans et al., 2009;Evans and Gardner, 2009). To further explore the role of P449, we investigated several changes to P449 here (**Table 1, right column**). First, we generated a P449A point mutant to proline to keep the 448-449 peptide bond in a *trans* configuration and provide more flexibility in the HI loop, both of which are normally associated with the WT conformation. In the [15]N/[1]H HSQC spectrum of this mutant, multiple peaks were observed for each amide residue (approximately three-fold more than in wildtype), none of which overlap with SLIP peaks, **Supplementary Figure S4**. This suggests that the added flexibility by the P449A mutation destabilizes the WT conformation and allows ARNT PAS-B to sample and adopt new stable conformations different from the previously-characterized SLIP state. In contrast, mutating P449 to either Ala or Gly in the Y456T background retained the two state WT:SLIP equilibrium but shifted it from the initial 50:50 value to 32:68 or 9:91, respectively. Interestingly, a P449A mutation in the SLIP-locked triple mutant (F444Q/F446A/Y456T) background, repopulated a small fraction of the WT conformation, reverting the WT:SLIP equilibrium to 7:93. These results suggest that the identity of the P449 residue impacts the WT:SLIP equilibrium, albeit in a role secondary to mutations in the β-sheet

residues. $^{15}$N/$^{1}$H HSQC spectra of the newly-generated ARNT PAS-B mutants P449G/Y456T and F444Q/F446A/P449A/Y456T are shown in **Supplementary Figure S5.**

Having found that mutations that increased flexibility at the HI loop proline could affect the relative populations of the WT and SLIP conformations, we wanted to test whether the length of the HI loop itself could contribute to the equilibrium. By integrating a TEV protease site into the middle of the HI loop (E-N-L-Y-F-Q, inserted between Y450 and S451), we could examine both how this insertion – and subsequent removal of the covalent linkage – would allow the Iβ-strand to slip while keeping the protein folded (**Figure 1a**, TEV insertion site on the HI loop shown in pink). NMR

characterization of Y456T with this loop insertion showed that the equilibrium between WT and the SLIP conformation was shifted to 20:80, suggesting that the addition of six residues in the HI loop favors slippage of the Iβ-strand and adoption of the SLIP conformation. Notably, after cutting with TEV protease, cleaving between the newly-introduced glutamine and S451, the equilibrium between the two conformations remained unchanged (**Supplementary Figure S5**). The sample was also applied to a MonoQ column to separate the two conformations, as we previously demonstrated (Evans et al., 2009);

immediately after injection onto the column, a precipitate formed, rendering it impossible to carry out the separation. We interpret the precipitation as arising from dissociation of the Iβ-strand peptide, as it is not observed either for this construct prior to TEV protease treatment nor for the wildtype protein. These results confirm that length in the HI loop influences the conformation of the Iβ-strand.

### 3.3     Mutations that promote SLIP conformation also affect the denatured state of ARNT PAS-B

Since the interconversion between WT and SLIP conformations proceeds through a chiefly unfolded transition state (Evans and Gardner, 2009;Xu et al., 2021), we hypothesized that the structure of the denatured state of the ARNT PAS-B may bias the WT:SLIP equilibrium. To address this possibility, we recorded triple-resonance NMR experiments on ARNT PAS-B wildtype, Y456T, and F444Q/F446A/Y456T in 8 M urea (a concentration at which most of the protein should be unfolded, **Supplementary Figure S6**) and used these data to assign the majority of backbone resonances for the three

variants. From these data, we predicted backbone amide N-H $S^2$ order parameters and secondary structure propensities in the unfolded state from measured chemical shifts using TALOS+ and SSP (Shen et al., 2009;Marsh et al., 2006), respectively. The predicted order parameters are consistent with all three proteins being highly flexible ensembles, with the majority of residues (85/119, 71%) displaying $S^2$ values < 0.6 and only 11 residues displaying $S^2$ values > 0.75. Consistent with this denatured state, the SSP algorithm predicted low amounts of residual secondary structure for all three proteins, with most

residues populating from 0 to 25% extended structure, **Figure 3**.

       The predicted order parameters are very similar for all three variants; the greatest difference is between residues 440 and 450 (Hβ-strand and HI loop), where the denatured F444Q/F446A/Y456T mutant (SLIP conformation) displays more order than the wildtype or Y456T proteins. In addition, the triple mutant has decreased extended structure propensity in the same region. Some of these differences may be due to the mutations present in this region; this is more likely for the

secondary structure propensity, which also detects a difference between the mutated and wildtype proteins in the region around residue 456 (also mutated). However, the $S^2$ values mainly differ in the Hβ-strand region, and also follow a rank order of (F444Q/F446A/Y456T)>Y456T>wildtype for residues 438-442. These findings show that the mutations that favor the SLIP conformation in the folded state also affect the residual structure in the unfolded state.

### 3.4 Compounds can preferentially bind one of the two ARNT PAS-B conformations

PAS domains can bind a variety of natural and artificial small molecule ligands (Henry and Crosson, 2011), many of which confer regulatory control. Combined with the observation of internal cavities within ARNT PAS-B (Guo et al. 2013), we tested whether small, artificial ligands could preferentially bind to either the WT or SLIP conformations of ARNT PAS-B. We did this by counterscreening previously-identified WT-binding ligands against the SLIP-locked F444Q/F446A/Y456T variant. The source of these ligands was a NMR-based screen of a library of 762 chemical fragments

(MW 203 Da on average, ±73 Da s.d.) which we previously used in similar screens of a variety of targets (Amezcua et al., 2002;Best et al., 2004;Guo et al., 2013;Scheuermann et al., 2009). For ARNT PAS-B, the screen started with initial $^{15}N/^1H$ HSQC spectra of $^{15}N$-labeled protein with pools of 5 fragments (1 mM each), which were ranked in order of the largest ligand-induced chemical shift and peak intensity perturbations compared to the apo protein (manually exempting pools which appeared to lead to protein denaturation). Pools showing substantial changes were then deconvoluted into individual

compounds which were independently added to identify ARNT PAS-B WT binders (**Figure 4a**). 18 hits were identified and were tested in separate titration experiments, ten of which with good solubility were considered to be ARNT PAS-B binders (Guo et al., 2013). To test whether these 10 ligands are specific to only the WT conformation of ARNT PAS-B, we collected $^{15}N/^1H$ HSQC spectra of the F444Q/F446A/Y456T variant (360 μM) mixed with the ligands (500 μM). Interestingly, none of the ligands showed chemical shift perturbations to the variant (example NMR spectra are shown in **Figure 4b**, rest of the

screen results are shown in **Supplementary Figure 7**), indicating all these compounds are WT specific.

   One of the tested compounds, KG-548 (5-(3,5-bis(trifluoromethyl)phenyl)tetrazole), exhibited slow exchange behaviour when titrated against WT ARNT PAS-B, with over 30 amide sites substantially broadened in the $^{15}N/^1H$ HSQC spectrum, (**Figure 4b, middle**). The affected peaks arose from residues that mostly localized to the β-sheet (**Figure 5a**). Interestingly, KG-548 was shown to be the strongest disruptor of ARNT PAS-B and CCC interaction in the previous study

(Guo et al., 2013), suggesting that its binding affects the β-sheet surface. More recently, through high-pressure NMR analysis, complemented by site-directed mutagenesis studies, we confirmed KG-548 to be a surface binding ligand, interacting hydrophobically with residues I364 and I458 on the external β-sheet surface of wildtype ARNT PAS-B (Gagné, 2020), which would explain the preferential binding, as I458 is flipped inward to the core of the protein in the SLIP conformation (**Figure 1b and Supplementary Figure 1b**).


### 3.5  KG-548 and KG-655 drive the conformational equilibrium towards the WT state

Since KG-548 preferentially binds the wildtype ARNT PAS-B, the addition of this compound to Y456T should drive the 50:50 (WT:SLIP) equilibrium towards the WT conformation. We titrated increasing concentrations of KG-548 (0 – 1000 μM) against 200 μM ARNT PAS-B Y456T, observing a shifted equilibrium of 77:23 (WT:SLIP) at the highest ligand concentration (**Figure 5b**). Due to the broadening of several peaks in the presence of the compound, we analyzed only those peaks that were not affected by ligand binding. Despite the difficulties posed by the broadening of peaks of interest (see Methods), we determined the binding affinity ($K_D = 684 \pm 33.1 \mu M$) and maximum binding ($B_{max} = 111 \pm 7.1$ μM, comparable to ~100 μM WT conformation in 200 μM ARNT PAS-B Y456T sample) of KG-548 for the WT conformation of ARNT PAS-B Y456T (**Figure 5c**).

Another ligand we characterized in detail was KG-655 (3,5-bis(trifluoromethyl)phenol), a fragment of KG-548, which was also shown to disrupt ARNT PAS-B binding to TACC3 (Guo et al., 2013). We previously showed that this ligand binds to two sites of wildtype ARNT PAS-B, both to the external side of the β-sheet surface and internally to the core of the domain (Gagné, 2020). We again performed titration experiments with increasing concentration of KG-655 (0 – 10 mM) and expectedly saw binding specificity towards the WT conformation, similar to KG-548 (**Supplementary Figure S8a**). Due to extensive chemical shift changes and broadening of many amide peaks in the presence of KG-655, we turned to $^{13}C/^{1}H$-HSQC titration experiments to extract binding affinities (see Methods). We chose to monitor the upfield-shifted L391 δ1 methyl signals because these peaks were well resolved in 1D and 2D NMR spectra and have been previously used to monitor the relative population changes between the two conformations (Evans and Gardner, 2009; Xu et al., 2021) (**Supplementary Figure S8b**). As an initial control, we calculated the dissociation constant ($K_d = 414 \pm 7.1$ μM) of KG-548 using this approach, comparable to the number reported above ($K_d = 684 \pm 33.1$ μM). With the same method, we extracted a $K_d$ of KG-655 ($1947 \pm 152$ μM) to ARNT PAS-B Y456T (**Supplementary Figure S8c**). Despite having extremely low binding affinity, the binding of KG-655 undoubtedly shifted the equilibrium towards the WT conformation. Interestingly, the surface binding of KG-655 to the WT conformation of ARNT PAS-B Y456T appeared to be abolished, leaving internal binding as the only binding mode for KG-655 (**Supplementary Figure S9**). We posit this is potentially due the loss of necessary interactions between the ligand and residue Y456. Taken together, these data confirm our previous findings that compounds can preferentially bind to a specific conformation of ARNT PAS-B and shift the equilibrium in the process. As noted above, we achieved population shifts with both surface- and core-binding ligands (KG-548 and KG-655, respectively).

The two examples above support a four-state model with the following states varying by conformation (WT, SLIP) and ligand binding (apo, bound), **Figure 5d**. In theory, the compound also binds the SLIP conformation, but our NMR data shows that the affinity between the two is minor, with no bound state visible even at the highest ligand concentration. Assuming that an invisible bound state would represent at most 10% of the protein in the SLIP conformation at the highest ligand concentration, we calculated a lower bound of 9 mM for the dissociation constant of KG-548 using Eq. 1, and even higher for KG-655. Taking KG-548 as an example, the value calculated allows us to estimate the equilibrium constant

between WT-bound and SLIP-bound states, as presented in **Figure 5d**. With the addition of compound to Y456T, the compound binds the WT conformation, creating a new WT bound state. This process depletes the WT unbound state, thereby driving the SLIP conformation towards WT to re-establish equilibrium.

## 4        Discussion

We have previously established the structural plasticity of the PAS-B domain of ARNT, which is highly sensitive to the side-chain at position 456, located in the Iβ-strand (Evans et al., 2009;Evans and Gardner, 2009;Xu et al., 2021). In the wildtype protein, this position is occupied by a tyrosine and the sidechain is solvent-exposed. Reducing the size of Y456 to a smaller sidechain enables ARNT PAS-B to enter an equilibrium between two stable conformations, the original WT state and a new SLIP state with a three-residue slip and inversion of the Iβ-strand which places this side-chain into the core of the protein. Here we extend our prior demonstrations of the ability of nearby mutations to influence the WT:SLIP equilibrium by evaluating the impact of environmental conditions, the HI loop, and small molecule binding on this interconversion. Taken together, these features give control over this uncovered flexibility intrinsic to this PAS domain.

It is clear that many PAS domains have significant intrinsic flexibility, particularly in the β-sheet surface and the final β-hairpin, consisting of HI loop linking the Hβ- and Iβ-strands. Supporting this idea, residues in the HI loop show intermediate exchange broadening in $^{15}N/^{1}H$ spectra of the wildtype ARNT PAS-B domain (Card et al., 2005), and when this domain is subjected to mechanical unfolding *in silico*, about 1/3 of the unfolding trajectories include a transient intermediate in which the C-terminal β-hairpin is unfolded (Gao et al., 2012). While this flexibility is expected to play a role in ligand entry to /exit from the core of a PAS domain, it is not evident in the static crystal structures of ligand-bound PAS domains. For example, only few residues in HIF-2α PAS-B show a significant (> 0.5 Å) backbone shifts between the apo and ligand-bound states (Scheuermann et al., 2015;Wu et al., 2015;Scheuermann et al., 2013). While the ARNT PAS-B mutations described here are not found naturally to the best of our knowledge, they appear to have uncovered hidden flexibility in sequences virtually identical to the wildtype protein. Our data here and previously suggest several structural determinants which enable the substantial β-strand rearrangement between the WT and SLIP conformations; similar characterization of the effects on dynamics using CPMG-type or other solution NMR experiments may provide additional mechanistic insights as well. Given the results shown in **Figure 3**, we suggest that attention on the unfolded (or nascently refolding) states are of particular interest in how these ultimately determine the relative ratio of different conformations (Evans and Gardner, 2009).

Our studies here and elsewhere (Evans et al., 2009;Evans and Gardner, 2009;Xu et al., 2021) on ARNT PAS-B Y456T comprise a toolkit for examining fragile protein folds and metamorphic proteins. Many of the approaches and lessons discussed here could be applicable to other similar biological systems, including some that have already been applied. For a simple example, the metamorphic protein IscU, which forms iron-sulfur clusters in *E. coli*, interconverts between two states in the process of carrying out its function; this interconversion requires *cis-trans* isomerizations of two proline peptide bonds

(Dai et al., 2012). This is reminiscent of the isomerization of the P448-N449 peptide bond that accompanies the switch from the WT to the SLIP conformation in ARNT PAS-B Y456T.

Critical to the ability for controlled switching between states is a small free energy difference between them. For ARNT PAS-B, we identify this as being -0.3 kcal/mol at 298 K – less than one hydrogen bond, illustrating that the loss or gain of only a few interactions would be enough to switch between the folds. The breakdown of enthalpic and entropic

contributions (ΔH: +6.8 kcal/mol, ΔS: +23.9 cal/mol-K) suggests that there are more favorable interactions in the SLIP state, but entropic considerations favor the WT state. Calculation of thermodynamic parameters of interconversion is relatively straightforward and could provide hints to the environmental triggers that favor one state over another in metamorphic proteins.

Fragile folds may interconvert through unfolded intermediates, or proceed through series of partially structured

transitions (Tyler et al., 2011;Zhao et al., 2016;Khatua et al., 2020). We previously reported that the interconversion between the WT and SLIP conformations in ARNT PAS-B Y456T goes through a mostly-unfolded intermediate that then refolds into two similarly stable folded states (Evans and Gardner, 2009;Xu et al., 2021). We find that the denatured state of the F444Q/F444A/Y456T mutant that favors the SLIP conformation has slightly greater order at the C-terminal end of the Hβ-strand. We posit that the Hβ-strand may form a folding core or nucleus in the F444Q/F446A/Y456T mutant that promotes

folding into the SLIP conformation; rigidity of the C-terminal end of the Hβ-strand may force the Iβ-strand to slip toward the C-terminus of the protein. It is also interesting and counterintuitive that, in this case, removing phenylalanine residues results in greater order in the unfolded state, since folding cores are often made up of such hydrophobic residues (Alexandrescu and Shortle, 1994;Buck et al., 1996;Klein-Seetharaman et al., 2002).

Changes in β-strand register have been noted in well-folded proteins, albeit rarely (Eigenbrot et al., 2001;Goldberg,

1998;Tuinstra et al., 2008;Wright and Scarsdale, 1995;Volkov et al., 2016). These findings prompted *in silico* studies to investigate such phenomena, where rearrangements of β-strand register have been observed through implicit and explicit solvent molecular dynamics (MD) simulations (Panteva et al., 2011;Li et al., 2007). However, during MD simulations of β-hairpin folding, it is often difficult to establish native β-strand register (Shao et al., 2013). Chong and co-workers proposed an "aromatic crawling" mechanism in which β-strand register is established via initial transient anchoring into hydrophobic

pockets, specifically mediated by phenylalanine residues (Panteva et al., 2011). While the situation in ARNT PAS-B is consistent with this proposed mechanism, it is not entirely the same, as F444, F446, and Y456 form a permanent hydrophobic cluster that clearly stabilizes the WT conformation. We suggest that this may reflect an underlying biophysical phenomenon: the importance of hydrophobic clusters on the "inside" face of the hairpin. In the case of proper folding, the formation of hydrophobic clusters on one face of the sheet drives the correct alignment of the β-strands relative to each other

and formation of cross-strand hydrogen bonds (Shao, 2015;Shao et al., 2013). This description may be applicable to the situation of ARNT PAS-B Y456T and F444Q/F446A/Y456T, as these two mutants progressively weaken a hydrophobic cluster (in this case found on the solvent-facing side of the β-sheet), thus allowing the Iβ-strand to slip during folding. In

fact, due to the hydrated cavity within ARNT PAS-B, it is an almost "inside-out" protein, with the solvent-exposed side of the β-sheet containing 12 polar and 11 non-polar residues, and 8 polar and 12 non-polar on the other side. The C-terminal β-hairpin of the HI loop is the most egregious example, with the Hβ-strand containing more aromatic residues on the solvent-exposed side and more polar residues on the other side and both sides of the Iβ-strand containing approximately the same numbers of polar and non-polar residues. This unconventional arrangement of sidechains on the ARNT PAS-B domain, likely due to its internal water-filled cavities and function as a protein binding site (Guo et al., 2013), may be the driving force behind its fragile fold. Similar studies of the unfolded states of other fragile protein folds may yield insights as well.

Work from our lab and others has established that PAS domains that contain surface grooves and interior cavities can bind ligands that modulate their interactions with other proteins (Guo et al., 2015;Henry and Crosson, 2011;Scheuermann et al., 2009;Wu et al., 2015;Gagné, 2020). Since the C-terminal β-hairpin is immediately adjacent to the interior cavity, we wondered if binding to a ligand could distinguish between the WT and SLIP conformations of ARNT PAS-B Y456T. Remarkably, we identified both surface- and core- binding ligands that selectively bound to one conformation over another. Two of these small molecules, KG-548 and KG-655, bind to the WT conformation with mid μM to low mM affinities and do not appreciably bind the SLIP conformation ($K_d > 9$ mM). Intriguingly, we have also shown that KG-548 and KG-655 binding to wildtype ARNT PAS-B can disrupt interactions with coiled-coil coactivators (Guo et al., 2015), suggesting that the conformational flexibility under study here may be relevant for biological interactions involving ARNT.

Finally, we emphasize our findings could complement other methods to manipulate conformational switches of proteins. For example, we have shown that equilibrium of protein conformations can also be pressure-dependent, mainly because of their volume and compressibility differences (Xu et al., 2021). Selective binding of ligands to one conformation over the other can affect such parameters, and change interconversion thermodynamics and kinetics, and therefore provide insights to the transition process. Other metamorphic proteins such as lymphotactin (Tuinstra et al., 2008) and IscU (Markley et al., 2013) have been shown to bind to specific binding partners in different folds. The N11L mutant of the Arc repressor, another engineered fragile fold, binds DNA and concomitantly drives the equilibrium to the native fold (Cordes et al., 2000). Our approach should be effective for characterizing these ligand interactions with fragile protein folds, as long as these proteins give rise to good NMR signals. Characterizing these interactions could help to reveal novel mechanisms for regulating biological activity by switching between two distinct structures and a means of controlling such switches (Ha and Loh, 2017).

## 5       Conclusions

We here characterize the thermodynamic components of a substantial change in protein conformation – a beta-strand register shift in a variant of the ARNT PAS-B domain – and demonstrate how it can be manipulated with a mix of environmental and sequence changes. Notably, we can exert control over the equilibrium using small molecule compounds

that preferentially bind one of the two states. Our findings suggest that it is possible to control the switch between the two structures using small molecules, providing a route for application to other proteins exhibiting fragile folds.

## Data availability

Backbone chemical shift assignments of the urea-denatured WT, Y456T, and F444Q/F446A/Y456T ARNT PAS-B proteins have been deposited at BMRB with the following accession codes: 50761 (wildtype), 50763 (Y456T), and 50762 (F444Q/F446A/Y456T). All other data are available upon request.

## Supplement

The Supplement contains the following information: Supplementary Figure S1 – S9.

## Author contributions

MRE, XX, ID, and KHG designed experimental approach and strategy. MRE, XX, and LPM performed the experiments. ID, XX, MRE, LPM, and KHG analyzed the data. XX, ID, MRE, KHG wrote the manuscript.

## Competing interests

The authors declare that they have no conflict of interest.

## Acknowledgements

We thank Prof. Robert Kaptein for the outstanding and inspirational research that he and his group contributed to the field of PAS domains and photoreceptors via their pioneering studies of the Photoactive Yellow Protein. We additionally thank Amy Zhou and other members of the Gardner laboratory for their constructive comments on this research.

## Financial support

This research was supported by NSF grant MCB 1818148 (to KHG) and NIH Grant T34 GM007639 (supporting LPM).

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

**Tables**

| Mutation | % WT conformation | % SLIP conformation | Mutation | % WT conformation | % SLIP conformation |
|---|---|---|---|---|---|
| *WT* | *>99* | *<1* | P449A | 14 | -[a] |
| *Y456A* | *>98* | *<2* | P449G/Y456T | 9 | 91 |
| *Y456S* | *81* | *19* | F444Q/F446A/P449A/Y456T | 7 | 93 |
| *Y456T* | *50* | *50* | Y456T + TEV (Uncleaved) | 19 | 81 |
| *P449A/Y456T* | *32* | *68* | Y456T + TEV (Cleaved) | 20 | 80 |
| *F444Q/Y456T* | *19* | *81* | | | |
| *F444Q/F446A/Y456T* | *<1* | *>99* | | | |

**Table 1: Effect of ARNT PAS-B mutations on WT:SLIP equilibrium values**

All ratios were established by $^{15}N/^{1}H$ HSQC spectra of each protein. Mutations listed in italics in left column have been previously published in (Evans et al., 2009). [a]Mutation P449A resulted in a complex mixture of WT and three additional non-SLIP conformations, populated to 60, 14, and 12% (detailed in **Supplementary Figure S4**).

## Figure Legends

**Figure 1 – ARNT PAS-B structure in the WT and SLIP conformations.** (a) Schematic of the ARNT PAS-B wildtype solution structure (PDB:1X0O (Card et al., 2005)), highlighting the F444, F446, and Y456 residues with space-filling representations. Internal cavities unique to the WT conformation (total volume: 105 Å$^3$) identified from a subsequent crystal structure (PDB:4EQ1 (Guo et al., 2013)) are shown in grey wireframe. A TEV protease site was inserted to the HI loop as part of this study, the insertion site in labelled in pink. (b) Schematics of the Iβ-strand residues in ARNT PAS-B wildtype (blue sticks) and F444Q/F446A/Y456T (orange sticks; PDB:2K7S (Evans et al., 2009)) structures. The external (solvent) face is above each strand and the internal side is below each strand. The location of the WT-specific internal cavities shown for reference in both WT and F444Q/F446A/Y456T.

**Figure 2 – Temperature dependence of WT:SLIP conformational equilibrium in ARNT PAS-B Y456T.** (a) Conformational preference of ARNT PAS-B Y456T at temperatures between 278 and 323 K. The equilibrium is linearly dependent on temperature (red line) between 278 and 303 K. Above 303 K, the equilibrium remains constant. (b) The data from panel (a) converted to ln($K_{eq}$) *versus* 1/T with the linear region (278-303 K) fit to a linear equation (red).

**Figure 3 – Backbone flexibility and secondary structure preference of ARNT PAS-B variants under urea denaturing conditions.** N-H S$^2$ order parameter predicted by TALOS+ (top) and residual secondary structure predicted by SSP (bottom; positive = helix, negative = strand) for ARNT PAS-B wildtype (black), Y456T (red), and F444Q/F446A/Y456T (blue) denatured in 8 M urea. The region of largest S$^2$ difference is shaded grey and the secondary structure elements of the folded wildtype protein are shown above the plot.

**Figure 4 – Screening small molecules binding to ARNT PAS-B WT and/or F444Q/F446A/Y456T variant.** (a) Schematic of the NMR-based screen used to identify small molecules binding to WT ARNT PAS-B (Guo et al., 2013). $^{15}$N/$^1$H HSQC spectra of 250 μM protein with mixtures of five compounds (1 mM each) were acquired and were scored for spectral differences between apo protein spectra (black) and those with compounds (red). Combinations which produced substantial peak shift or intensity changes (*e.g.* KG-545 to KG-549) were deconvoluted by acquiring spectra of individual protein/ligand mixtures, which were followed with NMR titration experiments for quantitative characterizations. (b) ARNT PAS-B binders from panel a) were counterscreened against the SLIP-conformation locked F444Q/F446A/Y456T variant (black spectra: apo protein; red spectra: protein/ligand mixture). Three examples (KG-279, KG-548, & KG-655) are shown, with additional examples in **Supplementary Figure S7**.

**Figure 5 – Characterization of KG-548 binding to ARNT PAS-B Y456T and impact on WT:SLIP conformation.** (a) Schematic of the ARNT PAS-B wildtype solution structure showing residues with $^{15}$N/$^1$H HSQC peaks broadened beyond detection by KG-548 as red sticks. These residues cluster around the β-sheet surface, specifically the central Iβ-strand. Location of E403, a residue distant from the ligand binding site and the cavities, is highlighted with green box (b) Overlay of $^{15}$N/$^1$H HSQC spectra of ARNT PAS-B Y456T in the absence (black) and presence of 1 mM KG-548 (red), showing selective binding of KG-548 to the WT conformation. Inset indicates the relative population change of residue E403 in the presence of KG-548. A zoomed-in view of the residue is also shown (green), with its location marked in panel a. (c) KG-548 binding to the WT conformation of ARNT PAS-B Y456T as monitored by volumes of ten $^{15}$N/$^1$H HSQC peaks (5 residue pairs) as a function of KG-548 concentration. Data were fit to Eq. 1 (red line) with best fit parameters as shown. (d) Diagram of a proposed four-state equilibrium of ARNT PAS-B Y456T in the presence of KG-548, with $K_d$ and $K_{eq}$ values as estimated in the text. Red ovals represent compound binding.

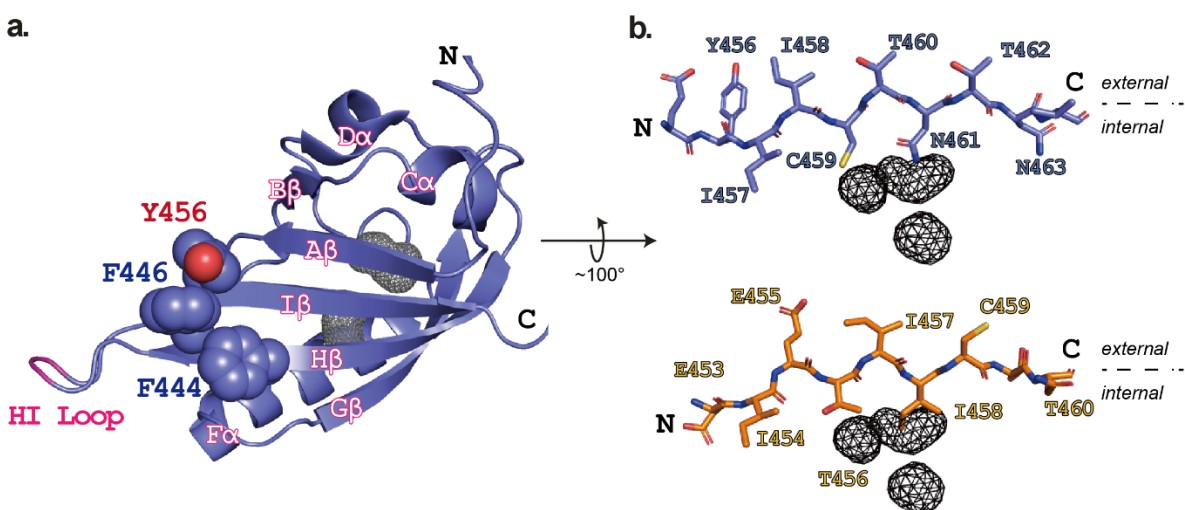

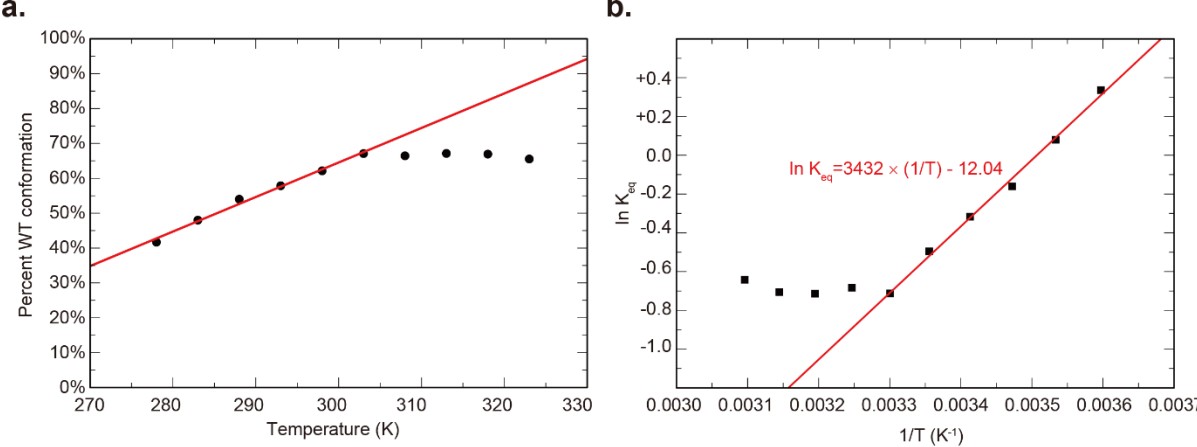

615

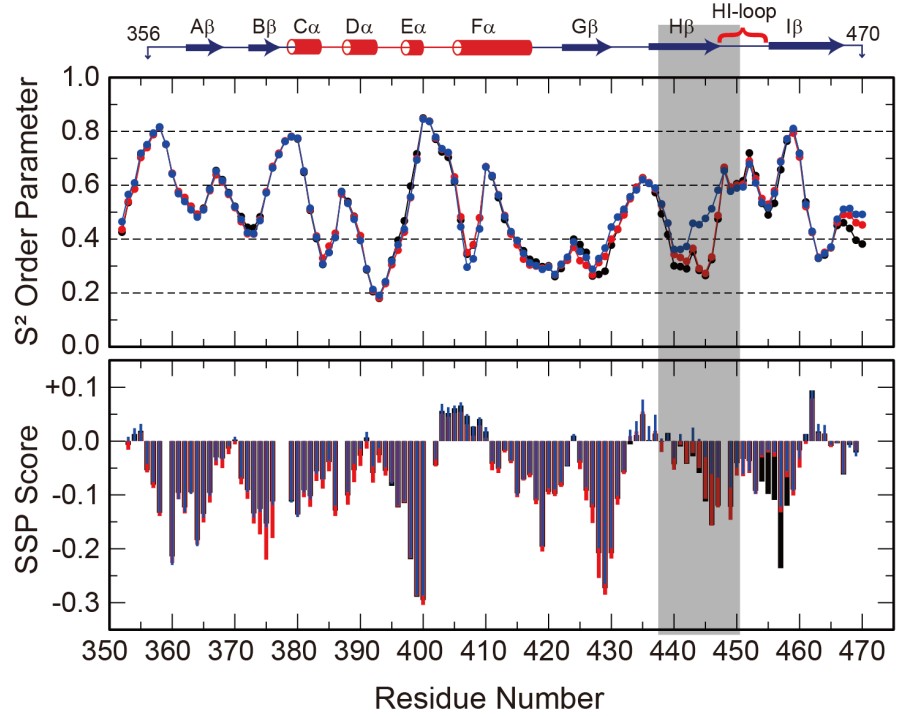

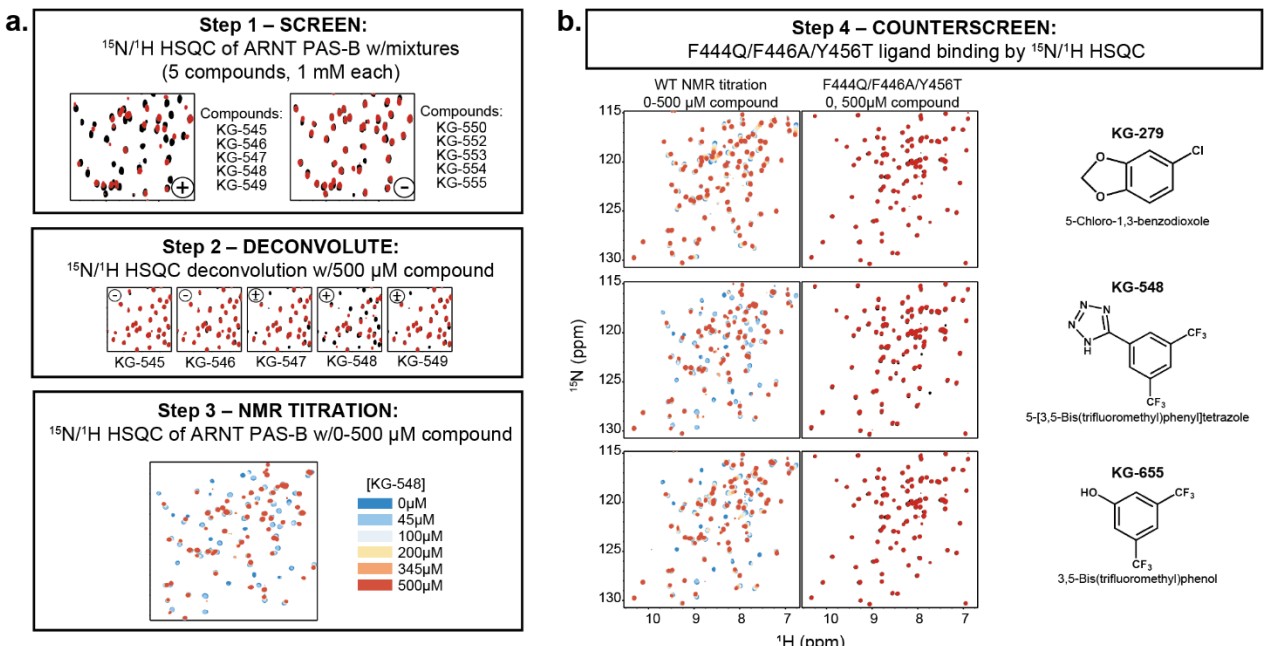

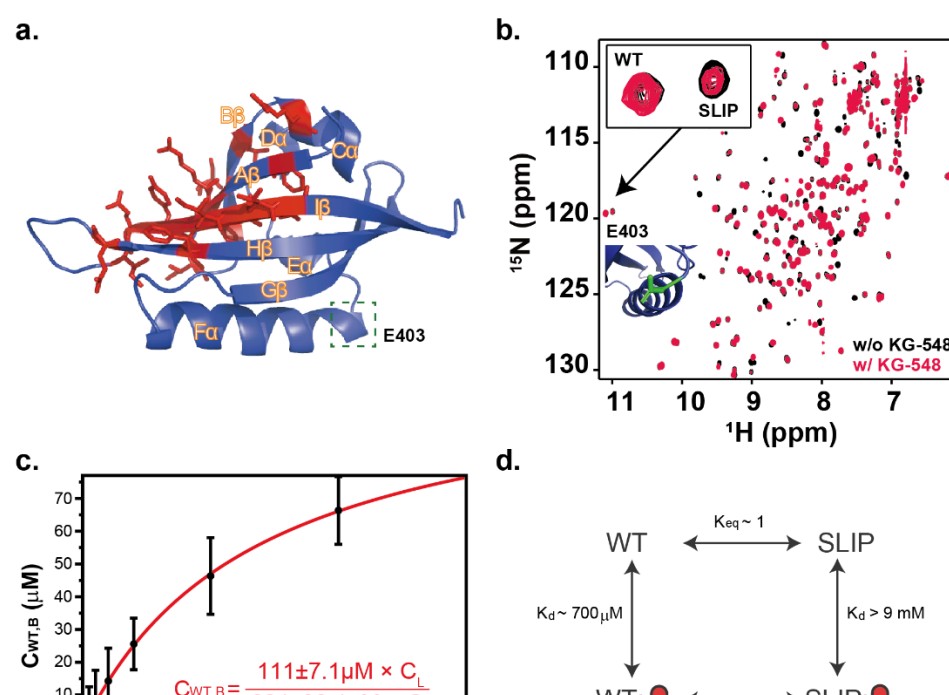

**a.**

**b.**

**c.**

$$C_{WT,B} = \frac{111 \pm 7.1 \mu M \times C_L}{684 \pm 33.1 \mu M + C_L}$$

**d.**