# Peer review of "Fragile protein folds: Sequence and environmental factors affecting the equilibrium of two interconverting, stably folded protein conformations"

_Magnetic Resonance, 2020_

## Referee Comment (RC2)

The review of this manuscript was performed jointly with Dr. Swati Balakrishnan and Dr. Yasiru Perera, who independently generated reviews that were subsequently discussed and edited by the team.  These are appended below.

**Review #1**

This paper uses a previously characterized interconversion of the ARNT PAS domain between two forms resulting from a single point mutation, to study various factors influencing this equilibrium. This study enhances our understanding of 'fragile fold' proteins and how they may be modulated. In general the paper is well written, and the inferences and conclusions drawn by the authors are borne out by the results of the experiments. One general concern is that the reasoning behind carrying out some experiments has not been clearly elaborated.

**Specific comments**
1) *Materials and methods section 2.5, page 6, line 170:* The authors mention they were unable to match the concentrations of the compounds KG-548 and KG-655 as additional DMSO would be required for the KG-548 sample. Is there some reason more DMSO cannot be added to the other sample in order to match the solvent? Some elaboration would be helpful here.
2) *Results section 3.1, page 8, line 190-195:* Varied individually, the effects of pH and salt concentration are minimal. Was any attempt made to try them in combination with each other, as this might help stabilize alternative conformations? Also why was the pH range chosen as 6.0-9.0? Is the protein unstable outside this range?
3) *Results section 3.2, page 9, line 214-226:* Adding residues to the HI loop region led to the conclusion that loop length plays a role in determining the conformation of the I-beta strand. However, adding a 6-residue TEV cleavage site to a 8 residue loop is a 75% increase in length. The residues added include bulky residues such as Tyr and Phe. It is unclear why multiple constructs were not created to incrementally increase the loop length instead, using residues such as Ala or Gly. Why insert a TEV cleavage site? It is also unclear why they chose to cleave off the I-beta strand, leading to the precipitation of the protein.
4) *Section 3.5, page 13, line 305-313:* The proposed model predicts Kd values assuming that an invisible bound state of ~10% of the protein is in the SLIP conformation. Could this invisible state be quantified more accurately by methods such as CEST, thereby making the calculation of the affinities of these compounds more accurate?

**Minor points**
1) *Materials and methods section 2.4, page 5, line 138:* chemical shift to be replaced by chemical shift perturbation
2) *Figure 5, page 12:* Panels need to renumbered, the text is referring to panel c as panel b. Panel d also has no error bars.
3) *Section 3.5, page 13, line 284:* 'compare to' to be replaced by 'compared to'

**Review #2**

This manuscript reports an investigation of the fragile fold of metamophoric protein PAS-B (Pre-ARNT-Sim) domain of human ARNT using solution NMR. ARNT PAS-B interconverts between two structural states, induced by the +3 residue slip of an internal Iβ strand. The two states are named WT and SLIP. The factors affecting the equilibrium of wild-type (WT) and mutant PAS-B were determined using multiple biophysical techniques. Solution NMR was used to measure conformational equilibrium constants and predict the order parameters of the denatured PAS-B. Preferential binding of ligands to the WT and SLIP states was monitored by a thermal shift assay and NMR. These studies concluded that this metamophoric equilibrium could be manipulated and potentially used in biological applications. The approach taken by the authors is sound and experiments appear to be well-designed. While the data supports the general conclusion there are several things to be addressed.

**Specific comments**
1. Results Page 7
$^{15}$N-$^1$H HSQC was used to characterize the equilibrirum ratio of WT:SLIP states. It would be useul to show the corresponding NMR spectra of two different states added in the supporting information as it would be more revealing to see the two populations in a single spectrum.
2. Results Page 7
The equilibrium has been monitored over a range of pH (6.0-9.0). It would be interesting to know if the authors tried acidic pH range or if they could speculate about the effects on the equilibrium at lower pH.
3. Results page 7
The authors have stated that mutating P449 to either Ala or Gly produced a shift in the equilibrium.  The result for Gly was expanded upon but no comparison was made with respect to the Ala mutant.
4. Results Page 8
The authors report changes in the equilibrium when the length of the HI loop increased. This was studied by adding a TEV cleavage site ENLYFQ. The authors note that the WT:SLIP equilibrium shifted to 20:80 concluding the effect of length. It is important to address the relative contribution of length versus sequence identity since no control experiments were performed. Are any structural changes induced by cleaving the HI loop at the TEV site?
5. Results Page 8
The manuscript would be strengthened by the addition of CPMG or another suitable NMR experiment to more completely characterize exchange dynamics of PAS-B and its mutants. Those data would inform loop motion and secondary structure changes associated with the conformational change. These results would be especially enlightening given the importance of HI loop in modulating the equilibrium. At the least it would be interesting to include this as a future direction.
6. Results Page 12. Figure 5d
No error analysis has been performed and the experiment seems to have been performed only once. If reporting a measure parameter reproducibility and precision should be addressed.

**Minor points**
1. Line 52 – add the bHLH abbreviation here.
2. Line 77 – Mention HI loop region in pdb structure (Figure 1a)
3. Line 184 – due to
5. Line 264/265 – Figure number 5b and 5c should be swapped.
6. Line 290 – Remove "," after (0-10mM)
7. Line 344 – Remove "actually"

---

## Author Response (AR1)

**Kevin H. Gardner, Ph.D.**
*Director, Structural Biology Initiative*
CUNY Advanced Science Research Center
Einstein Professor of Chemistry and Biochemistry, CCNY
kgardner@gc.cuny.edu
phone: +1 212-413-3220

February 21, 2021

Editors, *Magnetic Resonance*
*Uploaded via Magnetic Resonance website*

Dear Colleagues:

My co-authors and I thank Rolf Boelens as editor and the reviewers for their efforts evaluating our manuscript.  We are pleased that all three sets of reviewer remarks are both generally positive and contain a number of useful suggestions.  In response, we have edited the manuscript as detailed below (*reviewers' suggestions in italics*, our replies in plain text)*.

**Reviewer 1:**
*There are some areas that would benefit from additional discussion and/or data.*
*1. The thermal shift data (section 3.4) are confusing …*

We agree that the thermal shift data were confusing in some regards, including the mechanism of the destabilization of ARNT PAS-B seen across all binders.  This said, we felt that the novelty of ligand- and sequence-specific effects made these data still worth presenting, particularly as a primary screen for differential ligand binding that could be followed up in depth with NMR studies as we did with KG-548.  We've reconsidered that position given the reviewer's comments, and have decided to remove these data from the revised manuscript (leaving follow-up studies in a subsequent manuscript).

In lieu of this, we've rewritten Section 3.4 to include more data from a $^{15}N/^{1}H$-HSQC based fragment screen of ARNT PAS-B wildtype protein, showing that selected WT-binding compounds from this screen do not bind the mutant locked in the SLIP conformation.  While the NMR fragment screen has previously been reported (Guo *et al.*, *ACS Chem Biol* 2013), the counterscreen against the SLIP-locked mutant is novel to this work and demonstrates the generality of the conformational selectivity of ligand binding in a way that contributes to this manuscript.  We have added these data into new parts of Fig 4 and a new Supplementary Fig S7.

*2. Some point mutations (e.g. P449A) show multiple new HSQC peaks that are distinct from WT and SLIP peaks. The populations in Table 1 are reported as only WT and SLIP. Are the new peaks counted as SLIP or WT? How does one know which conformation to assign them to?*

We have added a footnote to the P449A entry in Table 1 indicating multiple conformations; all other entries in Table 1 generated only WT and SLIP peaks.  As we now detail further in Supplementary Figure 2 and accompanying legend, we observed a total of 4 sets of peaks, one assigned to WT (based on similarity in peak locations) and three non-WT/non-SLIP classes.  One of these was a dominant primary alternative conformation (60%), along with two minor ones of roughly equivalent population.  We have not done full chemical shift assignments to unambiguously establish the populations or structures of alternative species, which we view as out of the scope of the current manuscript.

[Figure]

*3. I recommend that the authors include data figures to support all stated observations. For example, the breakdown of compounds binding to WT vs. mutants (lines 258-260) and effects of loop insertion/cleavage (lines 218-222) could be illustrated using small HSQC panels showing which peaks shift and which don't.*

Data supporting these conclusions have been provided in new Supplementary Figs S5 and S7, along with some new components of Fig 4.

*4. The conclusions would be strengthened if the authors were to demonstrate that a compound that preferentially binds the SLIP conformation in fact shifted the population to that form, as they demonstrated for KG-548 and WT.*

We agree with the reviewer that having access to such SLIP-specific compound would be ideal, but presently our best validated compounds are WT binders resulting from the NMR-based fragment screen (including KG-548).  We see no reason *a priori* that a SLIP-specific compound cannot shift the Y456T equilibrium, but characterization of such compounds will be outside the scope of the current manuscript.

*5. Finally, the authors may wish to add a new reference for the lymphotactin metamorphic system [Dishman et al. (2021) Science 371, 86-90].*

Gladly added – this manuscript appeared after the first version of our paper was submitted.  We now cite this in the Introduction at one of the first references to the lymphotactin system.

**Reviewer 2a:**
*1) Materials and methods section 2.5: The authors mention they were unable to match the concentrations of the compounds KG-548 and KG-655 as additional DMSO would be required for the KG-548 sample. Is there some reason more DMSO cannot be added to the other sample in order to match the solvent?*

Two points influenced our maximum DMSO concentrations in the experiments: 1). Ligand solubility, and 2). 2% maximum DMSO concentration to avoid solvent-specific artifacts (e.g. destabilization during the noted 24 hr equilibration of protein:ligand).  In this specific instance, KG-548 is less soluble than KG-655, restricting the maximum concentrations applied to each ligand while still keeping [DMSO] < 2%.

*2) Results section 3.1: Varied individually, the effects of pH and salt concentration are minimal. Was any attempt made to try them in combination with each other, as this might help stabilize alternative conformations? Also why was the pH range chosen as 6.0- 9.0? Is the protein unstable outside this range?*

Our analyses of pH and salt were intended to be exploratory, e.g. quickly identifying any behavior reminiscent of the impact of such factors on the lymphotactin equilibrium, rather than a comprehensive survey.  As such, we opted to not test combinations of these effects.  The pH range of 6-9 was selected to survey conditions near physiological and to avoid crossing through the calculated pI (=5.6).

*3) Results section 3.2: Adding residues to the HI loop region led to the conclusion that loop length plays a role in determining the conformation of the I-beta strand. However, adding a 6-residue TEV cleavage site to a 8 residue loop is a 75% increase in length. The residues added include bulky residues such as Tyr and Phe. It is unclear why multiple constructs were not*

*created to incrementally increase the loop length instead, using residues such as Ala or Gly. Why insert a TEV cleavage site? It is also unclear why they chose to cleave off the I-beta strand, leading to the precipitation of the protein.*

We chose to insert a TEV cleavage site in the HI loop to test two hypotheses: 1). Lengthening the loop was predicted to favor the SLIP strand configuration due to strain, and 2). Interconversion would proceed through a chiefly-denatured intermediate that would only complete if the I-beta strand were covalently attached. Both hypotheses were proven correct, clearly for test 1) (e.g. Table 1 showing shifted SLIP population from Y456T) and more anecdotally for test 2) (e.g. precipitation of the protein over time). We have added additional data showing the effects of the loop insertion (and proteolysis) in Supplementary Figure S5.

*4) Section 3.5, page 13, line 305-313: The proposed model predicts Kd values assuming that an invisible bound state of ~10% of the protein is in the SLIP conformation. Could this invisible state be quantified more accurately by methods such as CEST, thereby making the calculation of the affinities of these compounds more accurate?*

This is an interesting suggestion, but one that we view as outside the scope of the current work. We underscore that the "10%" assumption is brought up in the context of ligand binding to different conformers; such binding would have to be extremely weak (e.g. mid-millimolar), raising questions about its specificity and utility for any practical application, and thus diminishing the importance of characterizing such a low-populated state.

*Minor points*
*1) Materials and methods section 2.4, page 5, line 138: chemical shift to be replaced by chemical shift perturbation*
*2) Figure 5, page 12: Panels need to renumbered, the text is referring to panel c as panel b. Panel d also has no error bars.*
*3) Section 3.5, page 13, line 284: 'compare to' to be replaced by 'compared to'*

All edits have been made in the text and figure as requested.

**Reviewer 2b:**
*1. Results Page 7: 15N-1H HSQC was used to characterize the equilibrium ratio of WT:SLIP states. It would be useful to show the corresponding NMR spectra of two different states added in the supporting information as it would be more revealing to see the two populations in a single spectrum.*

Such spectra are now provided in Supplementary Fig S2.

*2. Results Page 7: The equilibrium has been monitored over a range of pH (6.0-9.0). It would be interesting to know if the authors tried acidic pH range or if they could speculate about the effects on the equilibrium at lower pH.*

Please see comment above regarding our choice of pH range for this titration.

*3. Results page 7: The authors have stated that mutating P449 to either Ala or Gly produced a shift in the equilibrium. The result for Gly was expanded upon but no comparison was made with respect to the Ala mutant.*

Those data are now provided in Supplementary Figs S4 and S5, with additional brief commentary in the text.

*4. Results Page 8: The authors report changes in the equilibrium when the length of the HI loop increased. This was studied by adding a TEV cleavage site ENLYFQ. The authors note that the WT:SLIP equilibrium shifted to 20:80 concluding the effect of length. It is important to address the relative contribution of length versus sequence identity since no control experiments were performed. Are any structural changes induced by cleaving the HI loop at the TEV site?*

As noted in our response to Reviewer 2a above, our intention of these experiments was to provide a direct test of two hypotheses – that insertion would shift the WT:SLIP equilibrium, and that covalent linkage of the I-beta strand was important for stability of the protein given our models of interconversion via an unfolded intermediate. Both hypotheses were tested and confirmed, see above. We did not intend to present a more comprehensive analysis of the length and sequence-dependence of HI loop insertions in this forum, as this is a non-trivial endeavor in our opinion. We've added a brief note to the text to this effect.

*5. Results Page 8: The manuscript would be strengthened by the addition of CPMG or another suitable NMR experiment to more completely characterize exchange dynamics of PAS-B and its mutants. Those data would inform loop motion and secondary structure changes associated with the conformational change. These results would be especially enlightening given the importance of HI loop in modulating the equilibrium. At the least it would be interesting to include this as a future direction.*

We agree that the addition of these data may be of interest, although such work is well outside the scope of the current manuscript. We have added a brief note to this point in the text, emphasizing that these may be best done on the unfolded state (given the Figure 3 data) and with standard caveats of the sensitivity of certain experiments to motions on specific timescales.

*6. Results Page 12. Figure 5d: No error analysis has been performed and the experiment seems to have been performed only once. If reporting a measure parameter reproducibility and precision should be addressed.*

We have addressed this concern by adding error bars to these data (now Fig 5c), obtained by peak intensities from six NH peaks to provide independent measurements of the C(WT,B) parameter. We then used these values to guide the generation of 30 synthetic datasets to both obtain the fit dissociation constant and bootstrap our confidence intervals reported in the paper.

**Minor points**
1. Line 52 – add the bHLH abbreviation here.
2. Line 77 – Mention HI loop region in pdb structure (Figure 1a)
(points 3 and 4 truncated in referee's report)
5. Line 264/265 – Figure number 5b and 5c should be swapped.
6. Line 290 – Remove "," after (0-10mM)
7. Line 344 – Remove "actually"

All of these changes have been made.

**Reviewer 3:**

*1) Section 3.2, first paragraph: The authors state that "we could lock the protein into the SLIP conformation by adding the F444Q and F446A point mutations on the Hbeta strand to the Y456T background, enabling structural characterization of the SLIP conformation."*
*Of the two in silico studies of beta-sheet rearrangements that are mentioned in the manuscript, the Panteva et al., 2011 study has revealed that aromatic residues such as Phe appear to anchor into transient pockets as part of an "aromatic crawling" mechanism for shifting the beta-strand. Perhaps the authors could mention in the manuscript that the "locking" of the protein into the SLIP conformation via Phe->Ala mutations is consistent with the aromatic crawling mechanism that was previously observed in simulations? I think this is an interesting connection to make between experiment and simulation regarding the rearrangements of beta sheets.*

The aromatic crawling mechanism is consistent with the situation in ARNT PAS-B, but differs slightly: in the proposed mechanism, the hydrophobic cluster transiently anchors beta strands which then "crawl" to their correct register, whereas in ARNT PAS-B the cluster is stably formed and clearly contributes stabilizing energy to the WT fold. We have added a clarification to this effect in the Discussion.

*2) Since the beta-sheet shift is a key part of the interconversion between the WT and SLIP states, it would be helpful to include a schematic diagram that shows a) the hydrogen bonds in the beta-sheet that are being broken, and b) the amino acids (perhaps as beads with the one-letter code for amino acids) to highlight the alignment of nonpolar residues across the beta-sheet.*

We have provided a new Supplemental Fig S1 to show these points more clearly.

*3) Figure 1 is a bit confusing to me as the positions of residues F446 and F444 in panel a) are not indicated in panel b), which is supposed to be a 90-degree rotated view of panel b). Perhaps the suggested figure in my point 2) would be better as a panel b), particularly if the point is to highlight changes in hydrogen bonds that result from the beta-strand slip as well as which nonpolar residues are aligned across the beta-sheet.*

We have remade Figure 1 and provided a new Supplemental Fig S1 to address both comments.

We hope that these edits sufficiently address reviewer concerns and comments to satisfactorily proceed with publication; please contact me if I can provide any further information as the review proceeds.

Sincerely,

Kevin H. Gardner, Ph.D.
*Director, Structural Biology Initiative, CUNY Advanced Science Research Center*
*Einstein Professor of Chemistry and Biochemistry, City College of New York*